# Multicenter comparative analysis of local and aggregated data training strategies in COVID-19 outcome prediction with Machine learning

**Carine Savalli** [1,2] *, **Roberta Moreira Wichmann** [3], **Fabiano Barcellos Filho** [2], **Fernando Timoteo Fernandes** [4], **Alexandre Dias Porto Chiavegatto Filho** [2], on behalf of IACOV-BR Network [¶]

**1** Federal University of São Paulo, Department of Public Politics and Public Health, Santos, Brazil, **2** School of Public Health, University of São Paulo, São Paulo, Brazil, **3** Brazilian Institute of Education, Development and Research-IDP, Economics Graduate Program, Brasilia, Brazil, **4** FIAP–Faculdade de Informática e Administração Paulista São Paulo, Brazil

¶ Membership of the IACOV-BR Network is listed in the Acknowledgments.
* carine.savalli@unifesp.br

## Abstract

Machine learning (ML) is a promising tool in assisting clinical decision-making for improving diagnosis and prognosis, especially in developing regions. It is often used with large samples, aggregating data from different regions and hospitals. However, it is unclear how this affects predictions in local centers. This study aims to compare data aggregation strategies of several hospitals in Brazil with a local training strategy in each hospital to predict two COVID-19 outcomes: Intensive Care Unit admission (ICU) and mechanical ventilation use (MV). The study included 6,046 patients from 14 hospitals, with local sample sizes ranging from 47 to 1500 patients. Machine learning models were trained using extreme gradient boosting, lightGBM, and catboost for structured data. Seven data aggregation strategies based on hospital geographic regions were compared with local training, and the best strategy was determined by analyzing the area under the ROC curve (AUROC). SHAP (Shapley Additive exPlanations) values were used to assess the contribution of variables to predictions. Additionally, a metafeatures analysis examined how hospital characteristics influence the selection of the best strategy. The study found that the local training strategy was the most effective approach, in the case of ICU outcomes, for 11 of the 14 hospitals (79%), and, in the case of MV, for 10 hospitals (71%). Metafeatures analysis suggested that hospitals with smaller sample sizes generally performed better using an aggregated data strategy compared to local training. Our study brings to light an important concern about the impact of grouping data from different hospitals in predictive machine learning models. These findings contribute to the ongoing debate about the trade-off between increasing sample size and bringing together heterogeneous scenarios.

**Data Availability Statement:** The data comes from 14 distinct hospitals, and it is not publicly available as it contains information of patients in accordance with the Brazilian data protection law but are available on reasonable request from the Laboratory of Big Data and Predictive Analytics in Healthcare at the School of Public Health at the University of São Paulo (labdaps@usp.br). The Brazilian General Data Protection Law (LGPD) restricts the public release of deidentified records to protect individuals' privacy. A "reasonable request" typically refers to a request made for legitimate research purposes, where the requester can demonstrate a clear plan for maintaining data confidentiality and ethical standards. All the code written to develop the models can be found on https://github.com/labdaps/iacovbr_icu_mv_public.

**Funding:** This work was supported by National Council for Scientific and Technological Development (CNPq) under Grant Number 445020/2023-7. The author who received the grant was Alexandre Dias Porto Chiavegatto Filho. The funder did not play any role in the study design, data collection and analysis, decision to publish or preparation of the manuscript.

**Competing interests:** The authors have declared that no competing interests exist.

## Author summary

Machine learning (ML) in healthcare is often applied to large datasets, and a common strategy is to combine data from different regions and hospitals to increase sample sizes. In this study, we used ML models to predict two COVID-19-related outcomes: Intensive Care Unit admission (ICU) and mechanical ventilation (MV) use. We proposed different groupings of hospitals based on geographic regions and compared these with results obtained from individual hospitals (referred to as local training). The study found that local training generally provided more accurate predictions for the two COVID-19 outcomes. However, grouping hospitals for training prediction models was beneficial in cases where individual hospitals had few patients. We concluded that it is crucial to consider the local context before combining data from different centers with high data diversity.

## Introduction

The COVID-19 pandemic has posed unprecedented challenges to the world, with over 750 million confirmed cases and 6.9 million deaths attributed to COVID-19 [1]. While the peak of the crisis has passed, there are still long-term effects and severe clinical outcomes, specifically Intensive Care Unit admission (ICU) and the use for mechanical ventilation (MV), that need further investigation. Machine Learning (ML) has emerged as a promising tool for assisting in clinical decision-making, especially in the diagnosis and prognosis of COVID-19 [2,3].

During the peak of the pandemic, the overwhelming influx of patients into ICUs and the shortage of ventilators forced healthcare professionals to make difficult decisions regarding the prioritization of resources. By identifying patients at the highest risk of adverse outcomes, machine learning algorithms can facilitate the initiation of targeted preventive measures and optimize the allocation of physical resources within the healthcare system. Although there is a perceived gap in the application of advanced models to predict critical outcomes [4], a recent systematic review emphasized the potential of ML algorithms in addressing the complexities of COVID-19 [5].

Using a Brazilian multicenter cohort study, Wichmann et al. [6] applied machine learning algorithms to predict the risk of death from COVID-19 in 18 hospitals across the five regions of the country. The study compared eight different strategies for aggregating data to train the algorithms, and found that the strategy that yielded the best predictive performance for risk of death was training and testing the algorithms using only local hospital data.

The objective of this study is to investigate whether the observed trend of improved predictive performance with disaggregated data extends to other two COVID-19 related outcomes: the admission to the intensive care unit and use of mechanical ventilation. We hypothesized that hospitals with small sample sizes might benefit from data aggregation. Additionally, we aim to identify local characteristics that may be associated with more effective training strategies.

## Material and methods

A multicenter cohort study was conducted involving 16,236 adult patients who tested positive for COVID-19 across five regions of Brazil from March to August 2020. The study, encompassing 18 hospitals initially, focused on three primary outcomes: death, admission to the Intensive Care Unit (ICU), and the use of mechanical ventilation (MV). In the current study, we focused on the two last outcomes and included only adult patients (> 18 years) with a

positive RT-PCR (a standard laboratory diagnostic exam for COVID-19). Four hospitals were excluded from the current analysis as they either did not have ICU care and mechanical ventilation support, or only had ICU cases, as there was no local variation on the predicted outcomes. Consequently, the study included 14 hospitals from the five regions of Brazil: one from the Midwest, four from the Northeast, two from the North, four from the Southeast, and three from the South. The sample sizes at these hospitals ranged from 47 to 1,500 patients.

We trained three popular machine learning models for structured data: lightGBM, catboost, and XGBoost [7–9]. We included only variables routinely collected in all hospitals to train the algorithms: age, sex, heart rate, respiratory rate, systolic pressure, diastolic pressure, mean pressure, temperature, hemoglobin, platelets, hematocrit, red cells count, mean corpuscular hemoglobin (mch), red cell distribution width (rdw), mean corpuscular volume (mcv), leukocytes, neutrophil, lymphocytes, basophils, eosinophils, monocytes, and C-reactive protein. A single measure of these variables was collected in early hospital admission, within 24 hours before and 24 hours after the RT-PCR exam. Descriptive measures of quantitative predictors for each hospital are presented in S1 Table. The 22 variables were initially included in all models.

Box-plots were used to identify extreme values for continuous variables. If the maximum value was at least double the second maximum, it was replaced by missing to avoid non-plausible measures. This strategy helps maintain the overall quality and reliability of the data while addressing potential outliers effectively. Multinomial attributes were converted into dummy variables, creating separate variables for each category. Continuous attributes were normalized with z-score transformation. Variables exhibiting a correlation coefficient exceeding 0.90 were removed. Additionally, variables with over 90% missing values were also excluded. For the remaining variables with missing data, median imputation was initially applied.

We evaluated the method of multiple imputation using chained equations (MICE)[10], as it offers a balance between maintaining data integrity and computational efficiency, particularly given the large dataset and the relatively low percentage of missing values, but we found it did not improve the predictive performance of the models. Hyperparameters were optimized through Bayesian optimization using HyperOpt, coupled with 10-fold cross-validation. To address class imbalance, random oversampling technique was implemented in the training dataset, while the test dataset was left unchanged [11].

We compared eight data aggregation strategies for training (Table 1). The first strategy, called local training (Strategy 1), used data from the evaluated hospital for both training (70%) and testing (30%). We then applied seven data aggregations to compare with local training, and, for each one, we trained on 70% of the aggregated data and tested on 30% of the reference hospital data. For comparisons purposes, the 30% tested data set for each hospital was kept the same by fixing the randomization seed for all cases over the procedure. The aggregations proposed for training algorithms were: 70% of data from all hospitals, except the reference (Strategy 2); 70% of data from the same geographic region of the reference hospital, except the reference (Strategy 3); 70% of data from other geographic regions than the reference hospital (Strategy 4); 70% of reference hospital + 70% of data from all other hospitals (Strategy 5); 70% of reference hospital + same sample size selected from all other hospitals (Strategy 6); 70% of reference hospital + same sample size selected from hospitals of the same region (Strategy 7); 70% of reference hospital + same sample size selected from hospitals of other regions (Strategy 8). As in the Midwest there was only one hospital, analyses that aggregated hospitals within the same region for training purposes (Strategies 3 and 7) were not carried out for this hospital.

For each outcome (ICU admission and MV use), we considered all possible strategy-hospital combinations presented in Table 1, resulting in 14 hospitals x 8 strategies = 112 combinations. However, we were unable to analyze strategies 3 and 7 for the Midwest region, so the

**Table 1. Description of aggregation strategies of hospitals for training.** Abbreviations: SE: SouthEast, NE: NorthEast, MW: MidWest, S: South, N: North.

| Hospitals (reference) | Training Strategy | | | | | | | |
|---|---|---|---|---|---|---|---|---|
| | Strategy 1 local training | Strategy 2 | Strategy 3 | Strategy 4 | Strategy 5 | Strategy 6 | Strategy 7 | Strategy 8 |
| SouthEast2 (SE2) | 70% SE2 | 70% all other hospitals | 70% other SE | 70% NE +MW+S+N | 70% SE2 + 70% all other hospitals | 70%SE2 + same size selected from all other hospitals | 70%SE2 + same size selected from other SE | 70%SE2 + same size selected from NE +MW+S+N |
| SouthEast3 (SE3) | 70% SE3 | 70% all other hospitals | 70% other SE | 70% NE +MW+S+N | 70% SE3 + 70% all other hospitals | 70%SE3 + same size selected from all other hospitals | 70%SE3 + same size selected from other SE | 70%SE3 + same size selected from NE +MW+S+N |
| SouthEast5 (SE5) | 70% SE5 | 70% all other hospitals | 70% other SE | 70% NE +MW+S+N | 70% SE5 + 70% all other hospitals | 70%SE5 + same size selected from all other hospitals | 70%SE5 + same size selected from other SE | 70%SE5 + same size selected from NE +MW+S+N |
| SouthEast6 (SE6) | 70% SE6 | 70% all other hospitals | 70% other SE | 70% NE +MW+S+N | 70% SE6 + 70% all other hospitals | 70%SE6 + same size selected from all other hospitals | 70%SE6 + same size selected from other SE | 70%SE6 + same size selected from NE +MW+S+N |
| NorthEast1 (NE1) | 70% NE1 | 70% all other hospitals | 70% other NE | 70% SE +MW+S+N | 70% NE1 + 70% all other hospitals | 70%NE1 + same size selected from all other hospitals | 70%NE1 + same size selected from other NE | 70%NE1 + same size selected from SE+MW +S+N |
| NorthEast2 (NE2) | 70% NE2 | 70% all other hospitals | 70% other NE | 70% SE +MW+S+N | 70% NE2 + 70% all other hospitals | 70%NE2 + same size selected from all other hospitals | 70%NE2 + same size selected from other NE | 70%NE2 + same size selected from SE+MW +S+N |
| NorthEast3 (NE3) | 70% NE3 | 70% all other hospitals | 70% other NE | 70% SE +MW+S+N | 70% NE3 + 70% all other hospitals | 70%NE3 + same size selected from all other hospitals | 70%NE3 + same size selected from other NE | 70%NE3 + same size selected from SE+MW +S+N |
| NorthEast4 (NE4) | 70% NE4 | 70% all other hospitals | 70% other NE | 70% SE +MW+S+N | 70% NE4 + 70% all other hospitals | 70%NE4 + same size selected from all other hospitals | 70%NE4 + same size selected from other NE | 70%NE4 + same size selected from SE+MW +S+N |
| MidWest1 (MW1) | 70% MW1 | 70% all other hospitals | — | 70% SE+NE +S+N | 70%MW1 + 70% all other hospitals | 70%MW1 + same size selected from all other hospitals | — | 70%MW1 + same size selected from SE+NE +S+N |
| South1 (S1) | 70% S1 | 70% all other hospitals | 70% other S | 70% SE+NE +MW+N | 70% S1 + 70% all other hospitals | 70%S1 + same size selected from all other hospitals | 70%S1 + same size selected from other S | 70%S1 + same size selected from SE+NE+MW+N |
| South2 (S2) | 70% S2 | 70% all other hospitals | 70% other S | 70% SE+NE +MW+N | 70% S2 + 70% all other hospitals | 70%S2 + same size selected from all other hospitals | 70%S2 + same size selected from other S | 70%S2 + same size selected from SE+NE+MW+N |
| South3 (S3) | 70% S3 | 70% all other hospitals | 70% other S | 70% SE+NE +MW+N | 70% S3 + 70% all other hospitals | 70%S3 + same size selected from all other hospitals | 70%S3 + same size selected from other S | 70%S3 + same size selected from SE+NE+MW+N |
| North1 (N1) | 70% N1 | 70% all other hospitals | 70% other N | 70% SE+NE +MW+S | 70% N1 + 70% all other hospitals | 70%N1 + same size selected from all other hospitals | 70%N1 + same size selected from other N | 70%N1 + same size selected from SE+NE+MW+S |
| North2 (N2) | 70% N2 | 70% all other hospitals | 70% other N | 70% SE+NE +MW+S | 70% N2 + 70% all other hospitals | 70%N2 + same size selected from all other hospitals | 70%N2 + same size selected from other N | 70%N2 + same size selected from SE+NE +MW+S |

total number of strategy-hospital combinations was 110. We trained the three algorithms LightGBM, CatBoost, and XGBoost for each combination, leading to a total of 330 classifiers being trained. Initially, we selected the best algorithm for each strategy-hospital combination based on the area under the receiver operating characteristic curve (AUROC) obtained in the 30% test dataset of the reference hospital. Subsequently, we defined the best strategy for each hospital by comparing the AUROC of the eight strategies.

We used the binomial test to assess whether the proportion of hospitals favoring local training as the best strategy for ICU, MV, and death outcomes would be greater than 50% (one-sided test). Additionally, we calculated the 95% confidence interval for this proportion using the Clopper-Pearson exact method.

The contribution of variables to the prediction of outcomes was accessed by the SHAP (Shapley Additive exPlanations) values [12]. While these measures were initially obtained separately for the best model in each hospital, they were also utilized to assess the overall impact of variables that frequently emerged as significant predictors of the outcomes and the overlaps between the main predictors used for the optimal model at each hospital.

The 14 hospitals included in this study varied significantly in several aspects, including their geographic locations, sample sizes (ranging from 47 to 1,500 patients), the proportion of different outcomes, and the extent of missing data. Additionally, even though the variables collected were the same, there is an expected variation in their distributions. Therefore, we investigated whether these diverse characteristics among hospitals, called metafeatures, could explain why training models with local data might perform better than training with a larger volume of data and information by grouping other hospitals. The comparison among hospitals regarding the distribution of the 21 quantitative predictors was performed by calculating skewness, kurtosis, and the coefficient of variation and then summarizing them by the mean and standard deviation. For the three outcomes—ICU admission, mechanical ventilation (MV), and mortality—we then compared these metafeatures between two groups of hospitals. The first group comprised hospitals where a local training strategy was deemed most effective, while the second group included hospitals where an aggregation strategy proved superior. To compare the metafeatures between these groups, we employed the non-parametric Mann-Whitney test.

The study received ethical approval from the University of São Paulo's Institutional Review Board (IRB) under reference number 32872920.4.1001.5421, which included a waiver of consent. This approval also covered the utilization of data and collaboration with all hospitals participating in the IACOV-BR network.

## Results

The total sample consisted of 6,046 patients, with a mean age of 57.5 years (SD = 17.9), and distributed across the five regions, with 539 from Midwest (8.9%), 2389 from Northeast (39.5%), 294 from North (4.9%), 2129 from Southeast (35.2%), and 695 from South (11.5%).

In the overall sample, males were more prevalent (53.8% of the patients), while the most common racial background reported was white (65.4%).

Table 2 presents demographic data on age, sex, and race according to the ICU admission and the MV use, categorized by region. The proportion of patients who were admitted to the ICU varied from 22.1% in the Northeast to 95.4% in the Midwest, and the proportion of patients who used MV varied from 13.9% in the Northeast to 57.0% in Midwest, which highlights the diverse impact that the pandemic had on the different regions of the country. There is a tendency of patients being older among those admitted to the ICU and those who used MV. Male patients were more frequent in almost all regions and outcomes scenarios, but the

**Table 2. Descriptive statistics of the demographic's characteristics according to the ICU admission and the MV use, categorized by region.**

| | ICU | | MV | | |
| --- | --- | --- | --- | --- | --- |
| | **No** | **Yes** | **No** | **Yes** | **Total** |
| **Number of patients (n (%))** | | | | | |
| Midwest | 25 (4.6%) | 514 (95.4%) | 232 (43.0%) | 307 (57.0%) | 539 |
| Northeast | 1862 (77.9%) | 527 (22.1%) | 2058 (86.1%) | 331 (13.9%) | 2389 |
| North | 185 (62.9%) | 109 (37.1%) | 221 (75.2%) | 73 (24.8%) | 294 |
| Southeast | 727 (34.1%) | 1402 (65.9%) | 1267 (59.5%) | 862 (40.5%) | 2129 |
| South | 491 (70.1%) | 204 (29.4%) | 565 (81.3%) | 130 (18.7%) | 695 |
| **Total** | 3290 | 2756 | 4343 | 1703 | 6046 |
| **Age (mean ± std)** | | | | | |
| Midwest | 60.0 (17.2) | 57.9 (15.7) | 54.4 (15.4) | 60.7 (15.5) | 58.0 (15.8) |
| Northeast | 51.6 (18.4) | 65.4 (16.9) | 52.8 (18.7) | 66.2 (15.9) | 54.6 (19.0) |
| North | 52.8 (16.4) | 63.4 (15.2) | 54.1 (16.6) | 64.5 (14.8) | 56.7 (16.8) |
| Southeast | 57.0 (17.8) | 62.5 (15.6) | 59.6 (17.4) | 62.1 (15.2) | 60.6 (16.6) |
| South | 54.1 (17.6) | 67.2 (16.2) | 55.6 (17.8) | 68.0 (15.9) | 58.0 (18.2) |
| **Male (%)** | | | | | |
| Midwest | 64.0% | 62.1% | 65.1% | 59.9% | 62.2% |
| Northeast | 47.7% | 51.4% | 47.6% | 54.7% | 48.6% |
| North | 53.5% | 62.4% | 53.8% | 65.8% | 56.8% |
| Southeast | 49.7% | 59.6% | 53.0% | 60.9% | 56.2% |
| South | 55.2% | 62.3% | 56.3% | 61.5% | 57.3% |
| **Race—White (%)** | | | | | |
| Midwest | 4.0% | 15.8% | 3.0% | 24.4% | 15.2% |
| Northeast | 1.1% | 6.3% | 1.5% | 7.3% | 2.3% |
| North | 0.5% | 3.7% | 0.9% | 4.1% | 1.7% |
| Southeast | 53.4% | 47.5% | 50.1% | 48.6% | 49.5% |
| South | 81.3% | 74.0% | 81.1% | 70.8% | 79.1% |
| **Race—Black/Mixed/Asian (%)** | | | | | |
| Midwest | 20.0% | 31.4% | 11.6% | 45.4% | 30.8% |
| Northeast | 3.6% | 12.1% | 3.4% | 17.8% | 5.4% |
| North | 6.5% | 25.7% | 8.1% | 30.1% | 13.6% |
| Southeast | 24.0% | 28.6% | 24.9% | 30.2% | 27.0% |
| South | 2.4% | 1.0% | 2.1% | 1.5% | 2.1% |

Northeast presented a more balanced sample regarding biological sex. Among patients who provided information about self-declared race, there was a high proportion of white race in the Southeast and South regions, and this distribution reflects a similar tendency among patients with or without the two outcomes. Other races were grouped for the analyses, and they were more prevalent in the Midwest and Southeast.

Table 3 displays the AUROC results for the optimal strategy in each hospital, with scores exceeding 0.7 for almost all hospitals (the AUROCs for all strategies tested are in S2 and S3 Tables, respectively for ICU and MV outcomes). Predominantly, local training emerged as the most effective approach, being the superior strategy for 11 of the 14 hospitals (79%, p = 0.029, CI95% = [52%; 92%]) regarding ICU outcomes, and for 10 hospitals (71%, p = 0.089, CI95% = [45%; 88%]) for the MV outcomes. These results align with findings by Wichmann et al. (2023) regarding the risk of death, where local training was the best for 11 out of 18 hospitals included in that study (61%). In our study, local training yielded the best predictive performance for death risk in 9 of 14 hospitals (64%, p = 0.212, CI95% = [38%; 84%]).

**Table 3. AUROCs (AUCs) for the best aggregation strategies for training in each hospital for the three outcomes: the ICU admission, the use of mechanical ventilation (MV) and death (Wichmann et al. [6]).**

| Hospital* | n | ICU | | MV | | Death (Wichmann et al. [6]) | |
|---|---|---|---|---|---|---|---|
| | | Best AUC | Best strategy | Best AUC | Best strategy | Best AUC | Best strategy |
| Southeast—2 | 1500 | 0.799 | Local Training | 0.754 | Local Training | 0.79 | Local Training |
| Southeast—3 | 449 | 0.764 | Local Training | 0.865 | Local Training | 0.81 | Local Training |
| Southeast—5 | 124 | 0.667 | Local Training | 0.906 | Local Training | 0.98 | Local Training |
| Southeast—6 | 56 | 0.814 | Local Training | 0.885 | Local Training | 0.60 | Local Training |
| Northeast—1 | 1359 | 0.940 | Local Training | 0.947 | Local Training | 0.89 | Local Training |
| Northeast—2 | 845 | 0.666 | Local Training | 0.781 | Local Training | 0.92 | Local Training |
| Northeast—3 | 112 | 0.664 | Local Training | 0.779 | Local Training | 0.90 | Local Training |
| Northeast—4 | 73 | 0.744 | Strategy 2 | 0.908 | Strategy 7 | 0.70 | Strategy 4 |
| MidWest -1 | 539 | 0.673 | Local Training | 0.624 | Strategy 2 | 0.73 | Strategy 2 |
| South—1 | 456 | 0.748 | Local Training | 0.869 | Local Training | 0.90 | Local Training |
| South—2 | 148 | 0.987 | Local Training | 0.976 | Local Training | 0.98 | Local Training |
| South—3 | 91 | 0.770 | Strategy 6 | 0.722 | Strategy 2 | 0.85 | Strategy 3 |
| North—1 | 247 | 0.817 | Local Training | 0.885 | Local Training | 0.85 | Strategy 7 |
| North—2 | 47 | 0.909 | Strategy 4 | 0.796 | Strategy 2 | 0.90 | Strategy 5 |

*Hospitals named Southeast-1, Southeast-4, MidWest-2 and MidWest-3 in Wichmann et al. [6] were excluded for the current study.

The analysis of 42 cases across 14 hospitals, combining the three outcomes, showed that local training was the best in 30 out of 42 cases (71%, p = 0.004, CI95% = [56%; 83%]). This indicates that, in general, the proportion of cases where local training was the best strategy was significantly higher than 50%. Notably, when local training was chosen for mortality, it was consistently selected for both ICU and MV outcomes. In contrast, local training was not the preferred strategy for any of the three outcomes in three hospitals with smaller sample sizes—Northeast-4 (n = 73), South-3 (n = 91), and North-2 (n = 47). These hospitals exhibited no consensus on the optimal aggregation strategy, as preferences varied by outcome.

Across all strategies and hospitals for the two outcomes, the best-performing algorithm varied. LightGBM emerged as the top performer in 44.1% of cases, followed by extreme gradient boosting (XGBoost), which led in 35% of cases, and CatBoost in 20.9%.

Shapley values were computed for the best-performing model at each hospital, revealing variability in the key predictors across different institutions (S1 and S2 Figs). Despite this variability, certain variables like C-reactive protein (CRP), respiratory rate, and age consistently exhibited high Shapley values across most hospitals. Fig 1 provides a summary of the absolute Shapley values, highlighting the contribution of the top 10 variables that, on average, had the most significant impact on the predictions for each outcome.

The comparison between the group of hospitals where the local training provided better predictions and the group of hospitals that benefited from some aggregation strategy regarding the metafeatures of hospitals gives an insight into contexts where data aggregation can be advantageous and improve predictions (S4 Table presented metafeatures of all hospitals). The analysis indicated that hospitals with smaller sample sizes generally exhibited better performance using an aggregated data strategy compared to local training as we expected; this difference was statistically significant only for the ICU outcome, as shown in Table 4. Conversely, there was a trend where the local training strategy performed better in hospitals with unbalanced outcomes, which was confirmed statistically only for the MV outcome. No significant

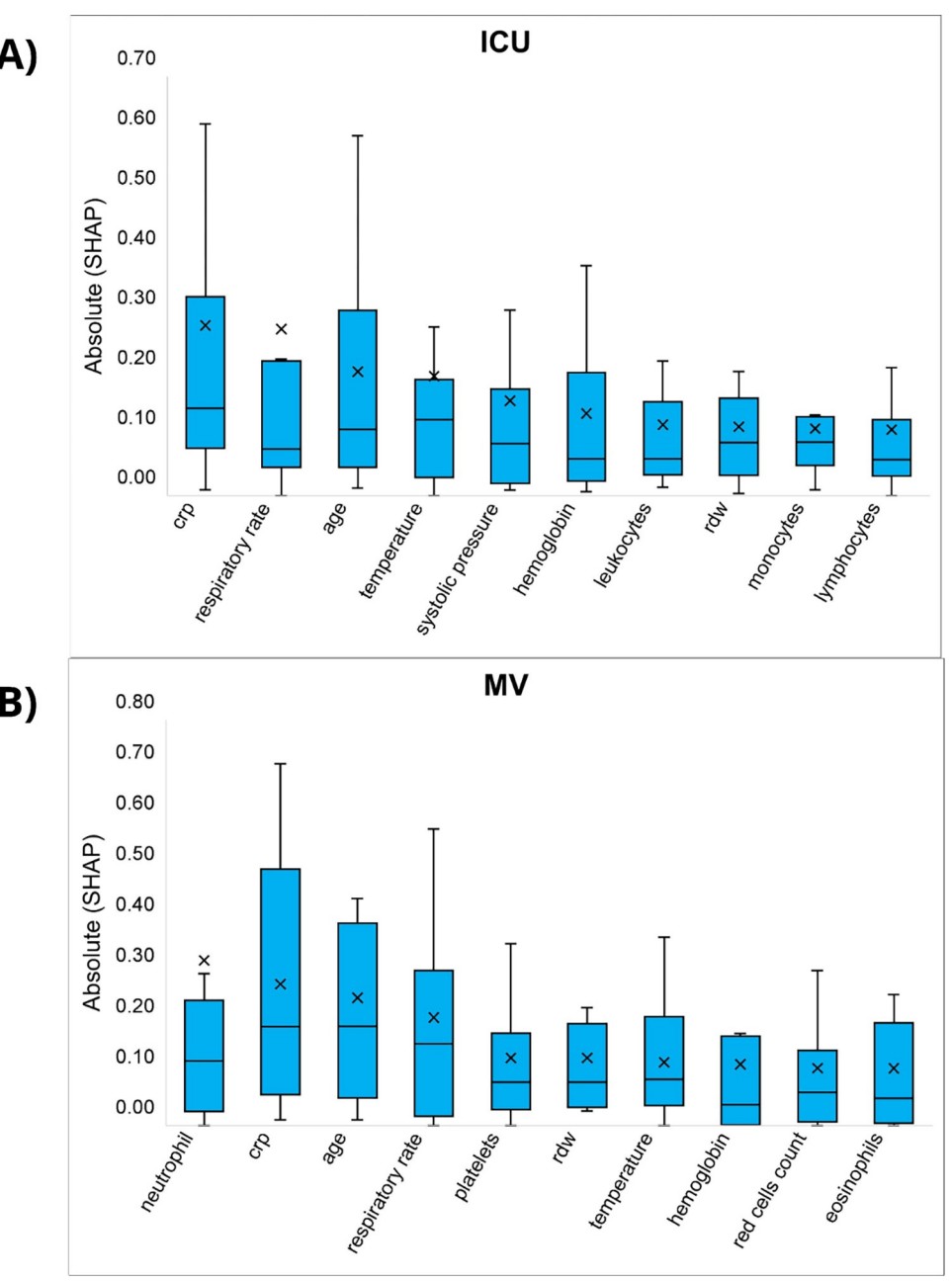

**Fig 1.** Box-plots of the absolute Shapley value obtained for the 14 hospitals for (A) ICU admission and (B) mechanical ventilation use. Each graph shows the 10 variables which, on average (for the 14 hospitals), presented high contributions to predicting the outcome.

differences were observed between the two groups of hospitals regarding the other metafeatures.

## Discussion

We found an overall tendency of better predictive performance for severe outcomes in hospitals with disaggregated data. For the three outcomes (ICU, MV and death), the best overall strategy was training and testing using only local hospital data. Our finding confirmed that the

**Table 4. Median and IQR (Interquartile range) of metafeatures and p-values of Mann-Whitney tests for comparing two groups of hospitals (local training versus some aggregation training), separately by type of outcome.**

| | ICU | | | MV | | | Death (Wichmann et al. [6]) | | |
|---|---|---|---|---|---|---|---|---|---|
| | Local training | Aggregation training | | Local training | Aggregation training | | Local training | Aggregation training | |
| **Metafeatures** | **11 hospitals** | **3 hospitals** | | **10 hospitals** | **4 hospitals** | | **9 hospitals** | **5 hospitals** | |
| | **Median (IQR)** | **Median (IQR)** | **p** | **Median (IQR)** | **Median (IQR)** | **p** | **Median (IQR)** | **Median (IQR)** | **p** |
| Sample size | 449 (556) | 73 (22) | **p = 0.022** | 348 (617.75) | 82 (136.5) | p = 0.142 | 449 (721) | 91 (174) | p = 0.190 |
| % of outcomes | 0.30 (0.41) | 0.51 (0.15) | p = 0.368 | 0.19 (0.18) | 0.50 (0.15) | **p = 0.024** | 0.14 (0.13) | 0.34 (0.09) | p = 0.060 |
| % of missings values | 0.22 (0.13) | 0.08 (0.10) | p = 0.126 | 0.22 (0.12) | 0.08 (0.06) | p = 0.054 | 0.22 (0.13) | 0.09 (0.14) | p = 0.190 |
| CV (average) | 0.57 (0.33) | 0.55 (0.18) | p = 0.885 | 0.54 (0.21) | 0.66 (0.34) | p = 0.539 | 0.57 (0.21) | 0.55 (0.37) | p = 0.898 |
| Skewness (average) | 1.67 (0.92) | 1.16 (0.29) | p = 0.368 | 1.46 (0.91) | 1.22 (0.77) | p = 0.945 | 1.67 (0.77) | 1.16 (0.59) | p = 0.364 |
| Skewness (std) | 3.10 (2.00) | 1.70 (0.60) | p = 0.060 | 2.86 (1.46) | 2.03 (1.52) | p = 0.374 | 3.10 (1.25) | 1.70 (0.77) | p = 0.112 |
| Kurtosis (average) | 15.04 (19.73) | 4.67 (2.72) | p = 0.060 | 13.86 (19.04) | 6.26 (13.91) | p = 0.374 | 15.04 (17.4) | 4.67 (3.39) | p = 0.112 |
| Kurtosis (std) | 24.69 (67.29) | 13.24 (6.63) | p = 0.088 | 24.04 (60.15) | 15.11 (29.24) | p = 0.454 | 24.69 (63.48) | 13.24 (8.84) | p = 0.147 |

CV: Coefficient of variation.

use of data from other hospitals to train algorithms for predictive purposes could increase data noise due to heterogeneity in hospital practices, and, consequently, decrease the predictive ability of the algorithms.

A distinct outcome emerged specifically for the three hospitals with small sample sizes. These hospitals, which benefited from the aggregated data, were not concentrated in any particular region of the country. Instead, their commonality was the small sample size, suggesting that sample size was an important factor influencing the results, rather than regional characteristics. Upon examining the metafeatures, it was observed that hospitals benefiting from data aggregation showed no significant differences in predictor distributions compared to those using local training. This finding supports the effectiveness of data aggregation for hospitals with limited data. Shapley values highlighted an overlap of certain variables that frequently appear as key predictors of the outcomes, although the importance of these variables varied among hospitals, reinforcing the value of adopting local models. Consistency across different clinical settings regarding the most important variables for predicting an outcome does not guarantee improved prediction by grouping the data.

Aggregating data from various regions and hospitals is a strategy frequently applied to increase the sample size for predictive models. Knight et al. [13], for example, developed a risk score to predict mortality in patients with COVID19, including several hospitals across England, Scotland, and Wales, stratifying into two geographical regions, and found that performance was robust across regions. Other studies that included data from different regions and countries found better performances as well [14, 15], however, it is important to discuss how this can impact predictions of local centers, especially in countries with high data heterogeneity.

Our findings offer practical insights and contribute to the ongoing debate about the benefits and drawbacks of aggregating data from various centers to predict diverse outcomes in healthcare. Increasing the size of the training sample does not uniformly enhance prediction accuracy. By increasing the volume of data from diverse clinical settings, there is a risk of introducing more noise due to measurement errors, inconsistencies, and inaccuracies, which

can manifest differently in each hospital. It is crucial to consider the local context before combining data from different centers. Brazil, with its vast geographical expanse and significant socioeconomic disparities, illustrates this point well. The country's five regions differ markedly in healthcare access and resources, leading to variability in hospital practices. This study underscores the importance of evaluating the cost-benefit of data aggregation from different hospitals. In scenarios characterized by high data heterogeneity, training a local model may be more advantageous.

The limitations of this study include regional imbalances in the sample distribution, potential inconsistencies in data collection procedures, and small sample sizes in some hospitals. Additionally, while we have observed an overall tendency for better predictive performance with disaggregated data, this observation requires further validation to confirm its robustness. We also recognize that there is a relationship between the three outcomes as some of the patients who died of COVID-19 underwent mechanical ventilation before, and/or went through the ICU as well. The criteria for ICU admission and for the use of mechanical ventilation may also vary across hospitals, which could increase data heterogeneity.

Nevertheless, we assert that predicting ICU admissions and mechanical ventilation (MV) use is as crucial as predicting mortality, and machine learning methods are effective for this purpose [5]. These predictions can assist clinicians in making informed decisions about treatment and resource allocation, potentially preventing patient deterioration. It is also important to recognize that the dynamics of COVID-19 have evolved since this cohort study was conducted. While the disease context has changed, the primary aim of this study was to evaluate the effectiveness of data aggregation in predicting various outcomes. Our findings highlight the need for careful consideration when combining data from different regions for predictive analyses, as regional disparities can significantly impact the outcomes. The analysis of the variations in sample size and the proportion of outcomes among different clinical settings can be a starting point for deciding whether to combine data from distinct locations.

## Supporting information

**S1 Table. Descriptive measures (mean, standard deviation and median) of predictors for each hospital.**
(DOCX)

**S2 Table. The AUROC achieved with the best algorithm (in parenthesis) in the test data set of each hospital for the eight aggregation strategies concerning the outcome ICU admission.** In green, the highest AUROC among the eight strategies for each hospital. Abbreviations: xgb: XGBoost, lgb: LightGBM, and cat: Catboost.
(DOCX)

**S3 Table. The AUROC achieved with the best algorithm (in parenthesis) in the test data set of each hospital for the eight aggregation strategies concerning the outcome outcome MV use.** In green, the highest AUROC among the eight strategies for each hospital. Abbreviations: xgb: XGBoost, lgb: LightGBM, and cat: Catboost.
(DOCX)

**S4 Table. Metafeatures of hospitals.**
(DOCX)

**S1 Fig. The most important predictors according to SHAP values for the best training strategy to predict the ICU outcome for each hospital (separately for 5 regions A-E).**
(DOCX)

**S2 Fig. The most important predictors according to SHAP values for the best training strategy to predict the mechanical ventilation (MV) outcome for each hospital (separately for 5 regions A-E).**
(DOCX)

## Acknowledgments

We would like to thank Sabrina Lima de Jesus for the figure design. We would like to thank the IACOV-BR Network. We would also like to thank all those people who somehow contributed to the progress of this research, in alphabetical order: Adriana Weinfeld Massaia; Alexandre Amaral; Ana Maria Pereira Rangel; Antônia Célia de Castro Alcantara; Bruna Donida; Bruno Mendes Carmon; Carisi Polanczyk; Carolina Zenilda Nicolao; Claiton Marques de Jesus; Denise Corrêa Nunes; Diana Almeida; Eduardo Menezes Lopes; Elias Bezerra Leite; Elimar Ponzzo Dutra Leal; Fernanda Arns de Castro; Fernanda Colares de Borba Netto; Flávia Araújo; Flávio Lúcio Pontes Ibiapina; Gerência de Ensino e pesquisa do Complexo Hospitalar da Universidade Federal do Ceará–EBSERH; Hospital Português da Bahia; Humberto Bolognini Tridapalli; Iasmin Luiza Leite; Laura Freitas de Faveri; Lena Claudia Maia Alencar; Luciane Kopittke; Luciano Hammes; Luiz Alberto Mattos; Marly Suzielly Miranda Silva; Mayara Rocha de Oliveira; Mohamed Parrini; Pablo Viana Stolz; Paloma Farina de Lima; Paulo Pitrez; Pollyana Bueno Siqueira; Rafaella Côrti Pessigatti; Raul José de Abreu Sturari Junior; Rodrigo Smania Garrastazu Almeida; Rogério Farias Bitencourt; Rubens Vasconcelos Barreto; Tatiane Lima Aguiar; Thyago Gregório Mota Ribeiro.

IACOV-BR Network
Ana Claudia Martins Ciconelle[5],
Ana Maria Espírito Santo de Brito[6],
Bruno Pereira Nunes[7],
Dárcia Lima e Silva[8],
Fernando Anschau[9,10],
Henrique de Castro Rodrigues[11],
Hermano Alexandre Lima Rocha[12,13],
João Conrado Bueno dos Reis[14],
Liane de Oliveira Cavalcante[15],
Liszt Palmeira de Oliveira[16],
Lorena Sofia dos Santos Andrade[17],
Luiz Antonio Nasi[18],
Marcelo de Maria Felix[19],
Marcelo Jenne Mimica[20],
Maria Elizete de Almeida Araujo[21],
Mariana Volpe Arnoni[22],
Rebeca Baiocchi Vianna[8],
Renan Magalhães Montenegro Junior[23],
Renata Vicente da Penha[24],
Rogério Nadin Vicente[25],
Ruchelli França de Lima[18],
Sandro Rodrigues Batista[26,27],
Silvia Ferreira Nunes[28,29],
Tássia Teles Santana de Macedo[30] &
Valesca Lôbo eSant'ana Nuno[31]
[5]Institute of Mathematics and Statistics, University of São Paulo, São Paulo, Brazil.

[6]Instituto de Medicina, Estudos e Desenvolvimento-IMED, São Paulo, Brazil.

[7]Universidade Federal de Pelotas-UFPel, Pelotas, Brazil.

[8]Hospital Santa Lúcia, Divinópolis, Brazil.

[9]Setor de Pesquisa da Gerência de Ensino e Pesquisa do Grupo Hospitalar Conceição, Porto Alegre, RS, Brazil.

[10]Programa de Pós-Graduação em Neurociências da Universidade Federal do Rio Grande do Sul, Porto Alegre, Brazil.

[11]Serviço de Epidemiologia e Avaliação/Direção Geral do HUCFF/UFRJ, Rio de Janeiro, Brazil.

[12]Unimed Fortaleza, Fortaleza, Ceará, Brazil.

[13]Departamento de Saúde Comunitária, Universidade Federal do Ceará, Fortaleza, Ceará, Brazil.

[14]Hospital São Francisco, Brasília, Brazil.

[15]Hospital Santa Julia de Manaus, Manaus, Brazil.

[16]Instituto Unimed-Rio, Universidade do Estado do Rio de Janeiro, Rio de Janeiro, Brazil.

[17]Universidade de Pernambuco-UPE/UEPB, Recife, Brazil.

[18]Hospital Moinhos de Vento, Porto Alegre, Brazil.

[19]InRad-Institute of Radiology, School of Medicine, University of São Paulo, São Paulo, Brazil.

[20]Departamento de Ciências Patológicas, Faculdade de Ciências Médicas da Santa Casa de São Paulo, São Paulo, Brazil.

[21]Federal University of Amazonas, University Hospital Getulio Vargas, Manaus, AM, Brazil.

[22] Serviço de Controle de Infecção Hospitalar Santa Casa de São Paulo, São Paulo, Brazil.

[23]Complexo Hospitalar da Universidade Federal do Ceará-EBSERH, Fortaleza, Brazil.

[24]Hospital Evangélico de Vila Velha, Vila Velha, Brazil.

[25]Hospital Santa Catarina de Blumenau, Blumenau, Brazil.

[26]Faculdade de Medicina, Universidade Federal de Goiás, Goiânia, Goiás, Brazil.

[27]Secretaria de Estado da Saúde de Goiás, Goiânia, Goiás, Brazil.

[28]Fundação Santa Casa de Misericórdia do Pará-FSCMP, Belém, Brazil.

[29]Mestrado Profissional em Gestão e Saúde na Amazônia, Belém, Brazil.

[30]Escola Bahiana de Medicina e Saúde Pública, Salvador, Brazil.

[31]Hospital Português da Bahia, Salvador, Brazil.

## Author Contributions

**Conceptualization:** Alexandre Dias Porto Chiavegatto Filho.

**Data curation:** Roberta Moreira Wichmann, Fernando Timoteo Fernandes.

**Formal analysis:** Carine Savalli, Fabiano Barcellos Filho.

**Investigation:** Carine Savalli, Roberta Moreira Wichmann, Fabiano Barcellos Filho, Fernando Timoteo Fernandes, Alexandre Dias Porto Chiavegatto Filho.

**Methodology:** Carine Savalli, Roberta Moreira Wichmann, Fabiano Barcellos Filho, Fernando Timoteo Fernandes, Alexandre Dias Porto Chiavegatto Filho.

**Project administration:** Alexandre Dias Porto Chiavegatto Filho.

**Resources:** Alexandre Dias Porto Chiavegatto Filho.

**Software:** Carine Savalli, Roberta Moreira Wichmann, Fabiano Barcellos Filho, Fernando Timoteo Fernandes.

**Supervision:** Alexandre Dias Porto Chiavegatto Filho.

**Writing – original draft:** Carine Savalli, Roberta Moreira Wichmann, Fabiano Barcellos Filho, Fernando Timoteo Fernandes, Alexandre Dias Porto Chiavegatto Filho.

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
