## [Decision Letter · Decision Letter 0]

23 Jul 2024

PDIG-D-24-00209

Multicenter comparative analysis of local and aggregated data training strategies in COVID-19 outcome prediction with Machine Learning

PLOS Digital Health

Dear Dr. SAVALLI,

Thank you for submitting your manuscript to PLOS Digital Health. After careful consideration, we feel that it has merit but does not fully meet PLOS Digital Health's publication criteria as it currently stands. Therefore, we invite you to submit a revised version of the manuscript that addresses the points raised during the review process.

Please submit your revised manuscript within 60 days Sep 21 2024 11:59PM. If you will need more time than this to complete your revisions, please reply to this message or contact the journal office at digitalhealth@plos.org. Please include the following items when submitting your revised manuscript:

We look forward to receiving your revised manuscript.

Kind regards,

Akhilanand Chaurasia

Section Editor

PLOS Digital Health

Journal Requirements:

Additional Editor Comments (if provided):

Dear Author,

Please provide the satisfactory rebuttal of queries raised by reviewers.

Reviewers' comments:

Reviewer's Responses to Questions

**Comments to the Author**

1. Does this manuscript meet PLOS Digital Health’s publication criteria? Is the manuscript technically sound, and do the data support the conclusions? The manuscript must describe methodologically and ethically rigorous research with conclusions that are appropriately drawn based on the data presented.

Reviewer #1: No

Reviewer #2: Partly

Reviewer #3: Yes

2. Has the statistical analysis been performed appropriately and rigorously?

Reviewer #1: I don't know

Reviewer #2: No

Reviewer #3: Yes

3. Have the authors made all data underlying the findings in their manuscript fully available (please refer to the Data Availability Statement at the start of the manuscript PDF file)?

Reviewer #1: No

Reviewer #2: No

Reviewer #3: No

4. Is the manuscript presented in an intelligible fashion and written in standard English?

Reviewer #1: Yes

Reviewer #2: No

Reviewer #3: Yes

5. Review Comments to the Author

Reviewer #1: As per my observation, this study has significance but couldn't be suitable for the publication, since by determining the stays in ICU and ventilation alone, could not assist much in medical significance towards the COVID treatment.

Reviewer #2: Motivation

----------

The paper poses and aims to answer a very important question for digital health: when predicting a patient's outcome from routinely collected medical data, is it better to restrict classifier training to particular hospitals/regions or not. The hypothesis is that variation across hospitals or regions may lead to poorer classifier performance when training data are aggregated.

Methodology

-----------

The paper tests the hypothesis using COVID-19 patient data from 14 Brazilian hospitals. The prediction problem has several input variables (actual number unclear) and 2 output variables (need for intensive care and need for medical ventilation).

The methodology considered eight data aggregation strategies, which include a baseline strategy (total aggregation) and others with varying levels of non-aggregation. Unfortunately this key aspect of the methodology is not well explained and I could not properly understand each strategy. The provided diagram (Fig 1) seemed more of a marketing image rather than explanatory.

Another methodological uncertainty was the use of oversampling for class rebalancing. The reader was directed to a 200-page text book for explanation. It is not clear what algorithm was used by the authors. It is also not clear why an off-the-shelf state-of-the-are method was not used. E.g., there are implementations of SMOTE (Chawla et al. “SMOTE: synthetic minority over-sampling technique,” Journal of artificial intelligence research, 321-357, 2002). Furthermore, many learning algorithms can operate directly with class-reweighting, rather than needing stochastic balancing methods. This was not discussed.

I also struggled to clearly understand the learning task given to the machine learning algorithms. The input variables that were considered, discarded and selected was somewhat unclear. Also, the entity of prediction was a patient admission, yet some input variables are measurements which may happen multiple times in an admission. Perhaps there are two or more values for some measurements?

A final methodological question exists about the choices of pre-processing to deal with extreme and missing values. The choices seem arbitrary with no justification (or reference to the research question).

Results

-------

The two principle independent variables are "hospital/region" and "aggregation strategy". The principle dependant variable is AUC. Unfortunately, nowhere in the paper is there a table showing AUC for each combination of hospital and strategy. Table 2 is close, but omits the results per strategy. Such results are crucial to the claim of the work. In particular, the reader cannot see if there is a statistical difference in AUC across strategy. Furthermore, the authors do not report statistical tests for the hypothesis that there is a difference.

Other important independent variables mentioned are "training set size" and statistical measures of data variation. These are somewhat systematically treated and reported (Table 3). However, the eight strategies are not made explicit, and conclusions are not clearly tied to these results.

The results of Shapley value analysis, while interesting, do not seem to directly relate to the posed research question. The relevance of Shapley values to the research question could be made clear (or the analysis omitted).

Conclusions

-----------

The authors seem to claim:

(1) "a consistent tendency of better predictive performance for severe outcomes in hospitals with disaggregated data"

(2) "the use of data from other hospitals to train algorithms for predictive purposes could increase data noise due to heterogeneity in hospital practices"

(3) hospitals with small training set sizes may benefit from data aggregation (see below),

(4) their "findings offer practical insights".

Claim (1) may be true and the authors may have collected the necessary data, however, the statistical validity of the claim is not established.

The results supporting claim (2) may be present in Table 3, however the authors could help the reader to see the connection.

Claim (3) was difficult to interpret (lines 207 to 211 on page 12). It would really help if the authors could clarify what they are claiming here, and make a clear connection to the presented results.

I am not convinced that claim (4) is established. In particular, I can't see where authors have explained to the reader what important consideration there are when deciding to aggregate data or not. Nor have they provided guidance on what may be appropriate aggregation methods.

The authors finish with, "Our findings highlight the need for careful consideration when combining data from different regions [...]." However, they do not state what those considerations are.

The authors also state, "These predictions assist clinicians in making informed decisions about treatment and resource allocation [...]." However, the cited work does not seem to make a good case for using such predictive analytics (like that presented by the authors) in clinical decision making. 

Language

--------

The prose of the submission I found was often ambiguous and lacked the precision expected of a scientific publication. Beyond me failing to understand the aggregation strategies, I also could not understand what was "the" RT-PCR exam (were there one or many, when did it happen, how was it used? what was the patient workflow?). There is much mention of "the reference hospital" but it was not clear which one was the reference hospital or what its purpose was in the method. Another minor example is that the authors state, "This study aims to compare data aggregation strategies [...] to predict COVID-19 outcomes such as [...] ICU and [...] MV." Why say "such as" when actually, it was "specifically" those two outcomes?

There seem to be a few typos when referring to figures and tables. E.g., line 78 on page 4, line 180 on page 10, and elsewhere.

When describing their work, the authors make a deal about the fact that data were extracted from 18 hospitals, but only data from 14 hospitals was analysed. This does not seem to be a big deal, but the reader is reminded of this fact multiple times from some reason.

Data Availability

-----------------

Finally, a comment regarding data availability. The authors claim they cannot release data due to "Brazilian data protection law". However, it is not clear what aspect of the cited law prevents release of deidentified records yet permit release given a "reasonable request". Furthermore, it is not clear what constitutes a "reasonable request".

Reviewer #3: Although the prevalance of COVID 19 has abrupted and the assesssed technology doesn't provide good results when large data set was used, the study could be quite helpful for the other future researchers who will work on the stated technology, It would be great insight for them, by referencing the present conducted research. This also increases the scope for modifications in the assessed technology for the long run, in order to tackle the bottlenecks of the present study.

6. PLOS authors have the option to publish the peer review history of their article (what does this mean?). If published, this will include your full peer review and any attached files.

**Do you want your identity to be public for this peer review?** For information about this choice, including consent withdrawal, please see our Privacy Policy.

Reviewer #1: No

Reviewer #2: No

Reviewer #3: No

---

## [Decision Letter · Decision Letter 1]

15 Oct 2024

PDIG-D-24-00209R1

Multicenter comparative analysis of local and aggregated data training strategies in COVID-19 outcome prediction with Machine Learning

PLOS Digital Health

Dear Dr. SAVALLI,

Thank you for submitting your manuscript to PLOS Digital Health. After careful consideration, we feel that it has merit but does not fully meet PLOS Digital Health's publication criteria as it currently stands. Therefore, we invite you to submit a revised version of the manuscript that addresses the points raised during the review process.

Please submit your revised manuscript within 30 days Nov 14 2024 11:59PM. If you will need more time than this to complete your revisions, please reply to this message or contact the journal office at digitalhealth@plos.org. Please include the following items when submitting your revised manuscript:

We look forward to receiving your revised manuscript.

Kind regards,

Dhiya Al-Jumeily OBE, PhD

Section Editor

PLOS Digital Health

Journal Requirements:

Additional Editor Comments (if provided):

Please address the reviewer's comments

Reviewers' comments:

Reviewer #1: All author comments have been addressed.

Reviewer #2: Overall

-------

The authors have done an excellent job of addressing most of the feedback provided in my initial review. However, some outstanding issues still remain that may be easy to resolve.

The authors have improved the justification for performing oversampling. However, it is still not clear to me which oversampling technique was used. The provided reference is a text book. Perhaps consider directing the reader to a particular page within the text that describes the method used.

It is still not clear to me if source variables were removed. The following two quotes seem to contradict.

1) "The 22 variables were kept in all models to maintain consistency across models for comparative purposes." [p5 line 86].

2) "Variables exhibiting a correlation coefficient exceeding 0.90 were removed. Additionally, variables with over 90% missing values were also excluded." [p5 line 93]

The description of the methodology is much improved (Table 1 really helps). However, I still struggle to fully understand it. After much analysis of the revised text I think I may understand it. I have summarised my understanding below. If this understanding is correct, I offer it as a possible framing for improving the description of the methodology.

I cannot find a statistical analysis of the primary hypotheses, namely that the "Local Training is better" and "Local Training is not better for small sample sizes." (I apologise if I have missed it.) The authors have provided all the data needed to perform such an analyses - which is excellent - but an analysis seems missing. I have performed my own analysis (below) and the results are favourable for the authors' claim. If this analysis is useful for the authors, I offer it as a possible template for their own analysis.

My Understanding of the Methodology

There are 14 hospitals (each is within 1 of 5 regions).

There are 8 strategies. For each strategy there are 14 trials [my terminology], where each trial is associated with one of the 14 hospitals, designated as "the reference hospital" for that trial.

Thus there are 8 x 14 = 112 possible strategy-hospital combinations. However, 2 combinations were not possible, explained below, i.e., there are only 110 strategy-hospital combinations. Each strategy-hospital is applied to 1 of 3 learners. There are two possible outcome variables. Thus there are 110 x 3 x 2 = 660 strategy-hospital-learner-outcome combinations all together.

For each hospital, the cases are partitioned so a [I assume random] 30% of cases are reserved for testing and the remaining 70% are used for training.

It is not clear if the train/test partitions were re-randomised or kept constant over the methodology. Specifically, are the same cases used for training/testing for each strategy-learner-outcome combination, or is data randomly repartitioned for each different strategy, for each different learner, for each outcome variable?

For each strategy-hospital-learner-outcome combination a classifier is trained. The training set is constructed according to a strategy and is relative to the trial's reference hospital. Then the classifier is applied to the reference hospital's testing data. 

Strategy 1: train on the reference hospital.

Strategy 2: train on all hospitals except the reference hospital.

Strategy 3: train on hospitals in the same region except the reference hospital.

Strategy 4: train on hospitals _not_ in the same region as the reference hospital.

Strategy 5: train on all hospitals, including the reference hospital.

Strategy 6: train on the reference hospital + a sample of the same size from all other hospitals.

Strategy 7: train on the reference hospital + a sample of the same size from other hospitals of the same region.

Strategy 8: train on the reference hospital + a sample of the same size from other hospitals _not_ in the same region as the reference hospital.

The Midwest region contained only one hospital, thus strategies 3 & 7 had no trial with that hospital as a reference.

For each strategy-hospital-learner-outcome combination, AUC was calculated. The classifier with the highest AUC was selected from each strategy-hospital-outcome.

The best strategy per hospital-outcome was found (modulo the two excluded strategy-hospital combinations). These 14 x 2 = 28 results are shown in Table 3.

My Statistical Analysis

Hypothesis: local training is better.

Null hypothesis: binomial distribution, 0.5 chance.

(ICU experiment)

Data: 11 of 14 (0.7857).

99% CI (Clopper-Pearson exact): 0.4108 -- 0.9743

p-value = 0.0287

(MV experiment)

Data: 10 of 14 (0.7143).

99% CI (Clopper-Pearson exact): 0.3421 -- 0.9474

p-value = 0.0898

(ICU + MV)

Data: 11 + 10 of 14 + 14, i.e. 21 of 28 hospital-outcome trials (0.7500).

99% CI (Clopper-Pearson exact): 0.4924 -- 0.9214

p-value < 0.00000001

(ICU + MV, with sample size >= 112, i.e., excluding small sample size)

Data: 10 + 9 of 10 + 10, i.e. 19 of 20 hospital-outcome trials (0.9500).

99% CI (Clopper-Pearson exact): 0.6829 -- 0.9997

p-value = 0.00002

-

---

## [Editor Report · Decision Letter 2]

10 Nov 2024

Multicenter comparative analysis of local and aggregated data training strategies in COVID-19 outcome prediction with Machine Learning

PDIG-D-24-00209R2

Dear Professor SAVALLI,

We are pleased to inform you that your manuscript 'Multicenter comparative analysis of local and aggregated data training strategies in COVID-19 outcome prediction with Machine Learning' has been provisionally accepted for publication in PLOS Digital Health.

Best regards,

Dhiya Al-Jumeily OBE, PhD

Section Editor

PLOS Digital Health

**Additional Comments:**

All comments have been addressed.